# How to transfer algorithmic reasoning knowledge to learn new algorithms?

**Louis-Pascal A. C. Xhonneux**[*]
Université de Montréal
Mila
xhonneul@mila.quebec

**Andreea Deac**
Université de Montréal
Mila
deacandr@mila.quebec

**Petar Veličković**
DeepMind, London UK
petarv@google.com

**Jian Tang**
HEC Montréal
Mila
jian.tang@hec.ca

## Abstract

Learning to execute algorithms is a fundamental problem that has been widely studied. Prior work [1] has shown that to enable systematic generalisation on graph algorithms it is critical to have access to the intermediate steps of the program/algorithm. In many reasoning tasks, where algorithmic-style reasoning is important, we only have access to the input and output examples. Thus, inspired by the success of pre-training on similar tasks or data in Natural Language Processing (NLP) and Computer Vision, we set out to study how we can transfer algorithmic reasoning knowledge. Specifically, we investigate how we can use algorithms for which we have access to the execution trace to learn to solve similar tasks for which we do not. We investigate two major classes of graph algorithms, parallel algorithms such as breadth-first search and Bellman-Ford and sequential greedy algorithms such as Prim and Dijkstra. Due to the fundamental differences between algorithmic reasoning knowledge and feature extractors such as used in Computer Vision or NLP, we hypothesise that standard transfer techniques will not be sufficient to achieve systematic generalisation. To investigate this empirically we create a dataset including 9 algorithms and 3 different graph types. We validate this empirically and show how instead multi-task learning can be used to achieve the transfer of algorithmic reasoning knowledge.

## 1 Introduction

Transfer learning [2] has been responsible for significant successes in multiple areas of machine learning, including Natural Language Processing (NLP) [3, 4] and Computer Vision (CV) [5]. Pre-training and reusing the learned weights, by freezing them as feature extractors, or fine-tuning from them as an initialisation, are common approaches to transfer in these domains. This has enabled successful learning on problems where data is limited in some form.

*Algorithmic reasoning* on graphs [1, 6] is a fundamental problem that has been studied under the assumption that we have access to the execution traces of the algorithms we want to learn. In practice, such as many real world reasoning tasks, this may not be true. Due to the limited data, we look to transfer learning to enable us to solve algorithmic tasks without intermediate steps. Specifically, we

---

[*]Corresponding author

35th Conference on Neural Information Processing Systems (NeurIPS 2021)

investigate how to transfer knowledge between similar *graph algorithms* in the setting where we have access to the execution traces for some (e.g. PRIM [7, 8]), but not others (e.g. DIJKSTRA [9]). This has not been explored before, but is important as many reasoning related fields care deeply about systematic generalisation, while only having input-output pairs to learn from. One such example is knowledge graph link prediction. There exists a reasoning process that can answer the question: what is the tail entity given the head entity and the relation. However, we do not have access to its execution trace, i.e. the step-by-step deductions of the reasoning process, but we do have input-output pair examples. We envision that the set-up studied and the direction proposed in this paper will help to learn neural networks that can solve reasoning-style problems by using other reasoning knowledge as an inductive bias. A slightly different, but related application may be when the data the graph algorithm needs to operate on is encoded in a high-level space. This is a scenario encountered in reinforcement learning and concurrent work Deac et al. [10] has already started investigating the process of pre-training an encoder with graph algorithms.

Veličković et al. [1] successfully trained graph neural networks (GNNs) [11, 12, 13, 14] to execute graph algorithms. Two key ingredients were access to the intermediate steps of the algorithm and algorithmic alignment [15]. Algorithmic alignment [15] refers to the concept of a computation structure—in our case neural networks—and the structure of an algorithm 'aligning'—in this paper graph algorithms. 'Alignment' means there exists a mapping between substructures of the computation architecture and simple substeps of the algorithm, where the substructures can 'easily' compute the corresponding substep they are mapped to—for some definition of easy, usually linear.

Extending this work, we enable solving tasks without access to the intermediate steps using other similar algorithms as an inductive bias. To do this we add several new algorithms. Further, since the algorithms studied in [1] were all expressible with only linear components in the GNN, we extend their architecture with a more expressive encoder to enable algorithmic alignment with more tasks.

One assumption we make is that the algorithms are similar (e.g. both PRIM and DIJKSTRA greedily select nodes from a priority queue). This shared algorithmic knowledge should serve as an inductive bias to enable a neural network to systematically generalise, even when learning only on input-output pairs. We hypothesise that due to differences between feature extraction and algorithmic reasoning, successful approaches to transfer as used in CV and NLP will provide only minimal improvements when transferring from one base algorithm to a target algorithm. While features across images may be similar, the features of algorithms differs (e.g. shortest distance in DIJKSTRA and lightest edge to the tree in PRIM), but are processed in a conceptually similar manner. Our intuition is that this conceptual relationship may not yield weights that are near each other in the weight space, thus making transfer difficult. We instead propose to use the base algorithms as inductive biases by training them with the intermediate steps along side training the target algorithm, for which we do not provide intermediate supervision. Our second hypothesis is that this will help systematic generalisation. We validate both hypotheses—transfer does not help, while multi-task learning helps—empirically.

The contributions of this paper are:

1. presenting a new benchmark for transferring algorithmic reasoning knowledge on graphs;
2. sampling the best trajectory to stabilise training when no execution traces are available;
3. show that standard transfer techniques fail to significantly improve systematic generalisation;
4. demonstrate how systematic generalisation can instead be improved on algorithmic tasks with multi-task learning.

## 2   Related Work

**Neural Execution**: Many papers have studied how to execute algorithms before [16, 17, 18, 19, 20]. With the rise of graph representation learning [21, 22, 23] and graph neural networks [11, 12, 13, 14], recent work has looked at executing graph algorithms [1, 6]. This line of work assumes access to the intermediate steps of the algorithm. Further, people have looked at teaching transformers to learn explicitly denoted subroutines and then learn how to combine them [24]. The key difference between our work and prior work is that we look at the setting of learning graph algorithms where we do not have access to the execution trace and we do not denote subroutines explicitly. Instead we propose to use similar algorithms as an inductive bias and look at how to enable this transfer of algorithmic knowledge.

**Transfer learning**: Transfer learning was originally proposed in the 1970s [25, 26]. Today, it is a common tool in NLP and Computer Vision to use pre-trained models and fine-tune them on the target task [2, 3, 5]. This is helpful when the target task is similar and the available data is limited in some way. Our work differs in two ways; firstly, we focus on algorithmic reasoning knowledge, which we hypothesis will require different techniques than reusing weights. Secondly, the data on the target task is partially missing specifically the execution trace of the algorithm is missing.

## 3   Background

This section gives background on the algorithms and graphs used for the experiments. To be precise we consider a graph to be a tuple $G = (V, E)$ with a set of vertices $V$ and a set of edges $E \subseteq V \times V^2$.

This section explains the algorithms studied in this paper in more detail. After that, we give a brief introduction into graph neural networks, the standard architecture paradigm for graph inputs.

### 3.1   Algorithms

We broadly study two classes of algorithmic reasoning: parallel and sequential reasoning. Specifically, the parallel algorithms (shown in Algorithm 1) studied in this paper exchange messages with their neighbours until an equilibrium is reached. Sequential algorithms, presented in Algorithm 2, greedily remove elements from a priority queue and updating the neighbouring nodes' keys.

| **Algorithm 1** Parallel | **Algorithm 2** Sequential |
|---|---|
| **Input:** graph $G$, weights $w$, source index $i$
initialise_nodes($G$.vertices, $i$)
**repeat**
    **for** $(u, v) \in G$.edges() **do**
        relax_edge($u, v, w$)
    **end for**
**until** none of the nodes change | **Input:** graph $G$, edge weights $w$, source node index $i$
initialise_nodes($G$.vertices, $i$)
$Q \leftarrow$ PriorityQueue($G$.vertices)
**repeat**
    $u \leftarrow Q$.pop_min()
    **for** $v \in G$.neighbours($u$) **do**
        relax_edge($u, v, w$)
    **end for**
**until** $Q$ is empty |

Different algorithms will implement their own `relax_edge` and `initialise_nodes` functions, however, the overall framework stays the same and can be learned by our GNN architecture (§ 4.1). For the sequential algorithms each node has a state feature indicating whether it has been removed from the priority queue, a key feature for the priority queue, and a pointer to the predecessor node.

We study the following algorithms:

PARALLEL

1. BREADTH-FIRST SEARCH (BFS)
2. BELLMAN-FORD
3. WIDEST PATH (PARALLEL)
4. MOST RELIABLE-PATH (PARALLEL)

SEQUENTIAL

1. PRIMS
2. DIJKSTRA
3. DEPTH-FIRST SEARCH (DFS)
4. WIDEST PATH (PARALLEL)
5. MOST RELIABLE-PATH (PARALLEL)

We give the pseudo-code for all algorithms in the supplementary. Note that the `relax_edge` function is increasingly difficult to learn for the neural network as you go down the list: PRIMS uses the edge weight as a key, thus the `relax_edge` function is the identity, for DIJKSTRA and DEPTH-FIRST SEARCH addition is necessary to compute the edge update, for WIDEST-PATH we need to take a maximum, which with a ReLU activation can still be done exactly, and for MOST RELIABLE-PATH the neural network needs to learn to approximate multiplication.[3]

---

[2]Throughout the paper we use graphs that are simple with weights.

[3]We note that the tasks are all linear time in the size of the graph. More challenging problems such as NP-hard problems are primarily more difficult because of their increased run-time and hence longer sequences allowing

## 3.2 Graph Neural Networks

A general framework to describe several graph neural network architecture is the message passing framework introduced by Gilmer et al. [13]. Such graph neural networks (GNNs) consist of three parts: a message function $M$, an update function $U$, and an aggregator $\bigoplus$. $M$ and $U$ are arbitrary neural networks. The high-level idea is

1. each node computes a message for its neighbours using $M$;
2. then each node aggregates the received messages using $\bigoplus$;
3. finally, each node updates its embedding using $U$.

The message function in this paper takes as input the sending and receiving nodes' representation as well as an encoding of the edge feature along which the message is sent. The aggregator chosen is the `max` aggregator applied element-wise as it was determined to work best by Veličković et al. [1] and provides algorithmic alignment [15] between the architecture and the tasks. The update function takes as input the node's previous embedding and aggregated messages.

## 3.3 Problem definition

We study learning graph algorithms that take in a graph $G$, a weight function $w : E \rightarrow \mathbb{R}$ that assigns a weight to each edge in the graph, and node features $X : V \rightarrow \mathbb{R}^k$. Algorithms compute an output $Y : V \rightarrow F^k$ and a predecessor $P : V \rightarrow V$ at each time step. This encompasses a large class of graph algorithms solving tasks such as reachability or shortest-path.

For instance, for a sequential algorithm the $i$th node's features $X_t[i]$ would be the `key` value and the `state` variable. Given $X_t$, the graph $G$, and the edge weights, the graph algorithm at each iteration returns the node features $Y_t$ and the predecessor for each $P_t$ (`pred` variable in pseudocode § 3.1). $Y_T$ and $P_T$ at the final step are considered the output of the algorithm. The intermediate steps refers to $X_t, Y_t, P_t \ \forall t \in \{1, \ldots, T-1\}$ after each iteration.

# 4 Methods

In this section, we first briefly present two architectures we will use for training and then discuss how existing algorithmic knowledge can be used to solve tasks when no execution trace is given.

## 4.1 Architecture

As our starting point we choose the encoder-processor-decoder architecture proposed in Veličković et al. [1]. The high-level idea is that the network encodes the current node state into a hidden embedding, on which a graph neural network unit, called the processor, is applied using the graph structure. The results are then decoded by the decoder, which does the prediction of the node features at the next time step (§ 3.1). The point of this architecture is to allow several algorithms to use the same processor architecture. A more detailed summary of the architecture is given below:

### 4.1.1 NeuralExecutor

The NeuralExecutor (NE) [1] uses an encoder-processor-decoder architecture. Let $X \in \mathbb{R}^{n \times k}$ be node states, where $n, k$ are the number of nodes and features, respectively. Each edge $(u, v)$ has a weight $w(u, v) \in \mathbb{R}$. The architecture keeps a hidden state for each node $H \in \mathbb{R}^{n \times l}$ with $l$ features, which is initialised to all zeros. The encoder $\mathcal{E}$ consists of a linear layer and computes a hidden embedding $\mathcal{E}(X_i, H_i) = Z_i$, where $i$ indicates the $i$th step in the computation. The processor $\mathcal{P}$ is message passing neural network (MPNN) with a `max` aggregator with linear message and update functions. The processor computes the new hidden state for each node $\mathcal{P}(H_i, A, w) = H_{i+1}$. Then, we have the decoder $\mathcal{D}(Z_i, H_{i+1}) = Y_{i+1}$ and predecessor predictor $\mathcal{S}(Z_i, H_{i+1}) = S_i$. Finally, a termination network $\sigma(\mathcal{T}(H_{i+1}))$ decides whether we should terminate or not.

---

for more accumulation of errors. As long as we have algorithmic alignment [15] between the architecture and the algorithm in question there is no additional challenge to NP-hard problems except the length of their sequence and hence more opportunities to introduce errors and propagate them.

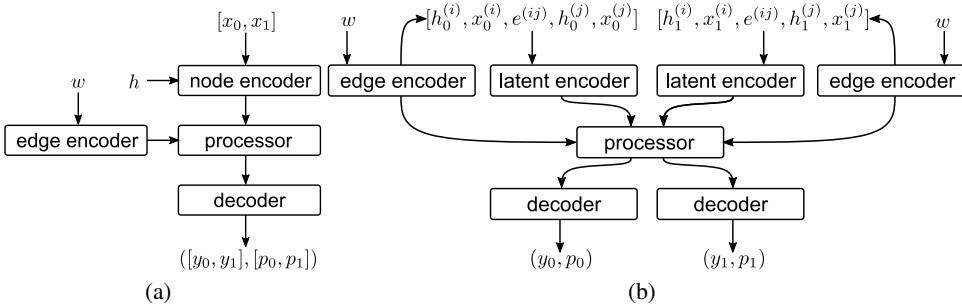

(a)                                                    (b)

Figure 1: $x_0, x_1$ are the node states of two algorithms respectively. $w$ are the edge weights. $p_0, p_1$ are the predecessor predictions for the two algorithms respectively and $y_0, y_1$ are the next node state predictions. $h_i$s are the previous hidden state kept by the network and $e^{(ij)}$ is the computed edge weight embedding. Superscript indices indicate a particular node. (a) Shows the original Neural Executor architecture when doing multi-task learning. (b) Shows the more expressive Neural Executor++ when doing multi-task learning. The key difference is that (b) forces a common way to operate on a hidden embedding space, while (a) focuses on achieving both at the same time.

We make minor changes to allow for better algorithmic alignment: we remove the `ReLU` activations from the processor and replace the termination layer with a processor and linear module[4].

### 4.1.2   NeuralExecutor++

Veličković et al. [1] showed positive transfer when learning algorithms in a multi-task setup with intermediate steps. They did so by concatenating together the node features and encoding them together into the same $h$ embedding (see Fig. 1). This has the advantage of giving strong guidance to the secondary algorithm learned in this multi-task set-up. However, ideally we are able to only use the base algorithm during training without having to use it at inference time introducing another failure mode. Thus, we propose to change the architecture as follows:

Since the node encoder is unable to learn the individual subroutines operating on edges, we replace it with a latent encoder for each task that operates on edges and for an edge $(i, j)$ takes in $[h^{(i)}, x^{(i)}, e^{(ij)}, h^{(j)}, x^{(j)}]$ (Fig. 1). The important difference is that each has task has its own encoder rather than all tasks sharing one encoder. The goal is to force the model to learn the shared subroutines in the processor only and the specialised subroutines in the latent encoder only (see Fig. 1).

Further, we change the latent encoder to consist of a linear and non-linear encoder in parallel that are added together. This should allow for algorithmic alignment with a much larger array of tasks (specifically WIDEST PATH), but comes at the cost that overfitting is more likely, which may hurt generalisation to larger graphs. This last problem has been avoided in prior work [1, 15] by having the neural network only learn linear components, which will not be able to overfit easily.[5]

### 4.2   Training and Loss functions

**Sequential algorithms (Seq)**: We used `softmax` for the prediction of the next node to be removed from the queue and `softmax` for the predecessor prediction, where we masked out all nodes except the neighbours and the node itself. We used a smooth $l_1$ loss with $\beta = 0.001$ for the prediction of the key of the selected node. Finally, we used `binary cross entropy` for termination prediction. We only update the node state of the chosen node, this helps with drift of the node state at test time.

**Parallel algorithms (Par)**: We used a smooth $l_1$ loss with $\beta = 0.001$ for the prediction of the key except for BFS where we used binary cross-entropy as the node state is either $0$ or $1$. In this setting, during teacher-forcing, we masked out nodes that are unreachable at a given time step.

---

[4]We remove the `ReLU` because it limits the ability of the max aggregator to minimise values by using negative inputs. The termination condition depends on whether the remaining nodes are still reachable hence an MPNN is a more appropriate than a linear layer.

[5]See the Supplementary for a more formal description.

**Teacher forcing (TF)**: We train the network to predict the next step given the ground truth inputs. At test time the networks prediction are used instead of the ground truth.

**No algorithm (NA)**: We train the network using only the loss on the final outputs, this means we change the `softmax` for predicting the next node to a `binary cross entropy` for sequential algorithms. The next inputs are those predicted by the network using `gumbel softmax` to predict the next node at each stage. The number of steps are given to the network in this scenario. Inspired by [27], we sample 10 trajectories using the best one for back-propagation, when using a `gumbel softmax`. The idea is that only the best loss is of interest at evaluation time and not the average loss. We show in the supplementary that this helps stabilise and improve training compared to taking the mean of the trajectories.

### 4.3   Standard transfer methods

We try three main approaches:

**Freeze weights**: This is a standard transfer technique, where we use the learned weights of the processor unit on a base algorithm $B$ for learning a target algorithm $T$. In this set-up we freeze the weights of the processor and only the encoder-decoder part of the architecture can be learned. We assume that the algorithm $B$ was learned with teacher forcing, while $T$ is learned with no-algorithm. We note that encoder-decoder weights cannot be reused because in general the input/output dimensions may differ from algorithm to algorithm.

**Fine-tune weights**: This is the same as the previous technique except that we do not freeze the processor weights and instead let them be changed by the gradient descent algorithm.

**2-Processors**: Again we assume we have access to the processor weights learned on a base algorithm $B$. This setting uses two processors in parallel whose outputs we sum together. One of the processor uses the learned processor weights and is frozen, while the other processor is free to learn the necessary changes and is randomly initialised. We suspect this will be the best transfer method as it does not forget the information given by the domain algorithm, but retains the flexibility to adapt the processor.

### 4.4   Transfer via multi-task learning

A stronger inductive bias is to train the base algorithm $B$, for which we have access to intermediate steps, together with the target algorithm $T$, for which we do not. This multi-task set-up highlights the difference between the original Neural Executor architecture proposed in Veličković et al. [1] and Neural Executor++ (see Fig. 1). The former simply concatenates the node features of $B$ and $T$ and forces them to share the hidden embedding. The latter allows different encoder-decoders and only shares the weights of the processor, i.e. each algorithm has its own hidden embedding. This second approach encourages the network to execute the shared subroutine in the processor and the individual subroutines in the encoder-decoder. The overall idea is that the base algorithm $B$ serves as an inductive bias as to how to structure the latent space and teach the processor how to evolve $T$.

### 4.5   Graphs

We study three kinds of graphs:

$$\text{i) Erdos-Renyi } p = \min\left(\frac{\log_2 |V|}{|V|}, 0.5\right), \text{ ii) Barabasi-Albert, iii) 2d-grid graphs.}$$

Erdos-Renyi graphs are random graphs where each possible edge has probability $p$ of being added to the graph. Barabasi-Albert graphs are power-law graphs with a few highly connected nodes and many dangling nodes. 2d-grid graphs are very regular graphs in an arbitrary 2d-grid shape. We chose these 3 classes of graphs because they represent some of the major possible differences between graphs. 2d-grid graphs are very regular and Veličković et al. [1] notes that very regular graphs tend to transfer poorly from or to random graphs. Erdos-Renyi graphs tend to be sparse, but highly likely to be connected. Barabasi-Albert graphs tend to be quite dense graphs with shorter average path-length than random graphs. As such these graph classes differ significantly from each other.

Table 1: Teacher forcing (seq.). NE++ worse performance on DIJKSTRA showing the downside of higher model capacity.

| Model | #Nodes | DIJKSTRA | | | MOST RELIABLE | | |
|---|---|---|---|---|---|---|---|
| | | Next node | Key | Predecessor | Next node | Key | Predecessor |
| NE | 20 | $0.018 \pm 0.004$ | $0.0367 \pm 0.01$ | $0.005 \pm 0.002$ | $0.238 \pm 0.05$ | $0.0358 \pm 0.02$ | $0.053 \pm 0.01$ |
| | 50 | $0.089 \pm 0.009$ | $0.569 \pm 0.7$ | $0.02 \pm 0.02$ | $0.557 \pm 0.08$ | $0.0697 \pm 0.05$ | $0.099 \pm 0.01$ |
| | 100 | $0.341 \pm 0.02$ | $4.79 \pm 6$ | $0.064 \pm 0.04$ | $0.763 \pm 0.07$ | $0.0924 \pm 0.06$ | $0.167 \pm 0.007$ |
| NE ++ | 20 | $0.008 \pm 0.003$ | $0.0108 \pm 0.004$ | $0.003 \pm 0.0008$ | $0.174 \pm 0.07$ | $0.0264 \pm 0.02$ | $0.047 \pm 0.03$ |
| | 50 | $0.35 \pm 0.03$ | $445 \pm 600$ | $0.211 \pm 0.02$ | $0.492 \pm 0.2$ | $0.0676 \pm 0.05$ | $0.112 \pm 0.05$ |
| | 100 | $0.729 \pm 0.01$ | $1.62e10 \pm 2e10$ | $0.522 \pm 0.07$ | $0.699 \pm 0.2$ | $0.0906 \pm 0.06$ | $0.171 \pm 0.06$ |

Table 2: No algorithm (seq.). Size-generalisation is lacking without intermediate steps, especially on the more difficult MOST RELIABLE.

| Model | #Nodes | DIJKSTRA | | MOST RELIABLE | |
|---|---|---|---|---|---|
| | | Key | Predecessor | Key | Predecessor |
| NE | 20 | $0.000104 \pm 7e\text{-}05$ | $0.023 \pm 0.01$ | $0.00829 \pm 0.001$ | $0.187 \pm 0.06$ |
| | 50 | $9.71e5 \pm 1e06$ | $0.121 \pm 0.1$ | $20 \pm 7$ | $0.897 \pm 0.08$ |
| | 100 | $3.34e16 \pm 5e16$ | $0.194 \pm 0.2$ | $1.32e6 \pm 4e5$ | $0.768 \pm 0.05$ |
| NE ++ | 20 | $6.4e\text{-}06 \pm 3e\text{-}06$ | $0.005 \pm 0.003$ | $0.000465 \pm 6e\text{-}05$ | $0.126 \pm 0.06$ |
| | 50 | $1.81 \pm 2$ | $0.149 \pm 0.06$ | $125 \pm 70$ | $0.566 \pm 0.06$ |
| | 100 | $8.54e3 \pm 1e4$ | $0.388 \pm 0.05$ | $2.8e13 \pm 4e13$ | $0.76 \pm 0.05$ |

For all graphs, we generate edge weights that are uniformly between $[0.2, 1.0]$, this range prevents key values such as shortest path from becoming too extreme[6].

# 5 Experiments

For all experiments we use 5,000 graphs of each type (Erdos-Renyi (ER), Barabasi-Albert (BA), 2d-Grids (2d-G)) with 20 nodes each. We train using ADAM [28] with a learning rate of $0.0005$, a batch size of $64$, and use early stopping with a patience of $10$ to prevent overfitting. We test on graphs size 20, 50, and 100 nodes. The hidden embedding size is set to 32 except for NE++ for multi-task experiments, where it is 16 to account for the additional expressivity of having several encoders. Each experiment was executed on a V100 GPU in less than 5 hours for the longest experiment.

We measure the average performance over all 3 graph types at evaluation separately and present the average with standard deviation in the main paper. Large standard deviation may arise due to the extreme difference between random graphs of type ER or BA versus 2d-G graphs.[7]

## 5.1 Metrics

**Sequential algorithms**: *Predecessor (Pred.)* error rate is the most important measure as to whether a task has been successfully completed as it gives us the path predicted by the network. *Next node (Next)* error rate measures whether the next node is the correct one to pick. *Key* accuracy measures whether the key of the picked node is correct, measured in mean squared error. *Next node* are indicative of whether the correct algorithm is being executed, while *Pred.* and *Key* primarily serves to indicate the correctness of the solutions found. Lower is always better.

**Parallel algorithms:** *Key* accuracy measures the node features mean squared error for all algorithms except BFS, where it is measured in accuracy as the node feature is a binary choice between 0 and 1. *Predecessor (Pred.)* accuracy measures the accuracy of predicting the predecessor node.

# 6 Results and Discussion

For the sequential algorithms we study transfer from PRIM to DIJKSTRA and from WIDEST PATH to MOST RELIABLE PATH, for parallel Algorithms we study transfer from BFS to BELLMAN-FORD and from WIDEST PATH to MOST RELIABLE PATH.

---

[6]All code to generate data and train models will be released upon acceptance with an MIT license.

[7]See the Supplementary for a more detailed explanation.

Table 3: Transfer to no algorithm (seq.). Pre-trained on PRIM and WIDEST, respectively. Classic transfer learning fails to provide size-generalisation and often performs worse than no transfer.

| Model | #Nodes | DIJKSTRA | | MOST RELIABLE | |
| | | Key | Predecessor | Key | Predecessor |
| --- | --- | --- | --- | --- | --- |
| | 20 | $0.014 \pm 0.005$ | $0.081 \pm 0.03$ | $0.0235 \pm 0.003$ | $0.248 \pm 0.06$ |
| NE Freeze | 50 | $122 \pm 200$ | $0.597 \pm 0.09$ | $3.91e5 \pm 3e4$ | $0.864 \pm 0.009$ |
| | 100 | $3.18e6 \pm 4e6$ | $0.607 \pm 0.1$ | $5.06e15 \pm 7e14$ | $0.761 \pm 0.04$ |
| | 20 | $0.0021 \pm 0.0009$ | $0.05 \pm 0.02$ | $0.036 \pm 0.005$ | $0.227 \pm 0.07$ |
| NE Fine-tune | 50 | $1.11e3 \pm 1e3$ | $0.241 \pm 0.09$ | $2e3 \pm 200$ | $0.636 \pm 0.1$ |
| | 100 | $5.93e7 \pm 8e7$ | $0.388 \pm 0.1$ | $1.06e9 \pm 1e8$ | $0.709 \pm 0.02$ |
| | 20 | $0.00136 \pm 0.0005$ | $0.06 \pm 0.03$ | $0.0163 \pm 0.003$ | $0.231 \pm 0.08$ |
| NE 2-Processor | 50 | $14.2 \pm 20$ | $0.162 \pm 0.06$ | $205 \pm 30$ | $0.749 \pm 0.04$ |
| | 100 | $1.04e4 \pm 1e4$ | $0.305 \pm 0.07$ | $1.22e10 \pm 4e9$ | $0.815 \pm 0.03$ |
| | 20 | $0.00136 \pm 0.001$ | $0.063 \pm 0.02$ | $0.00687 \pm 0.0008$ | $0.199 \pm 0.09$ |
| NE++ Freeze | 50 | $42.6 \pm 50$ | $0.841 \pm 0.04$ | $465 \pm 100$ | $0.58 \pm 0.1$ |
| | 100 | $262 \pm 400$ | $0.895 \pm 0.07$ | $8.06e7 \pm 3e7$ | $0.672 \pm 0.08$ |
| | 20 | $0.000414 \pm 0.0003$ | $0.034 \pm 0.02$ | $0.00669 \pm 0.002$ | $0.191 \pm 0.07$ |
| NE++ Fine-tune | 50 | $13.5 \pm 20$ | $0.962 \pm 0.03$ | $1.79e5 \pm 2e5$ | $0.757 \pm 0.05$ |
| | 100 | $2.51e4 \pm 4e4$ | $0.962 \pm 0.04$ | $8.17e14 \pm 5e14$ | $0.774 \pm 0.05$ |
| | 20 | $0.00443 \pm 0.002$ | $0.06 \pm 0.04$ | $0.0022 \pm 0.0003$ | $0.154 \pm 0.05$ |
| NE++ 2-Processor | 50 | $16.6 \pm 20$ | $0.356 \pm 0.1$ | $306 \pm 70$ | $0.644 \pm 0.03$ |
| | 100 | $4.66e3 \pm 6e3$ | $0.779 \pm 0.02$ | $3.58e6 \pm 9e5$ | $0.791 \pm 0.02$ |

Table 4: Multi-task (seq.): Using PRIM and WIDEST as inductive bias, respectively. Multi-task learning shows good generalisation, especially on the Key metric for DIJKSTRA and on Predecessor for MOST RELIABLE.

| Model | #Nodes | DIJKSTRA | | MOST RELIABLE | |
| | | Key | Predecessor | Key | Predecessor |
| --- | --- | --- | --- | --- | --- |
| | 20 | $0.00362 \pm 0.0005$ | $0.042 \pm 0.01$ | $0.207 \pm 0.03$ | $0.452 \pm 0.09$ |
| NE | 50 | $11.5 \pm 2$ | $0.134 \pm 0.1$ | $1.85 \pm 0.5$ | $0.501 \pm 0.06$ |
| | 100 | $126 \pm 30$ | $0.303 \pm 0.3$ | $6.47 \pm 3$ | $0.597 \pm 0.01$ |
| | 20 | $0.000178 \pm 8e\text{-}05$ | $0.019 \pm 0.009$ | $0.00279 \pm 0.0004$ | $0.166 \pm 0.07$ |
| NE ++ | 50 | $0.413 \pm 0.4$ | $0.161 \pm 0.1$ | $0.199 \pm 0.3$ | $0.185 \pm 0.009$ |
| | 100 | $2.91 \pm 3$ | $0.282 \pm 0.2$ | $0.843 \pm 1$ | $0.267 \pm 0.1$ |

## 6.1 Sequential

The first experiments establish baselines in terms of achievable performance given the intermediate steps and trained with teacher-forcing (Tab. 1). We run each algorithm separately. Next we establish the performance in the no-algorithm setting (Tab. 2), i.e. what is achievable without intermediate supervision.

**Expressivity can harm systematic generalisation**: Firstly, we note that the additional expressivity of the NE++ (§ 4.1.2) seems to hurt systematic generalisation even with the large amount of data available (Tab. 1) as we can see on DIJKSTRA. On MOST RELIABLE both do equally well on *Key* and *Pred.*, but looking at *Next node* we can see that NE++ does better in simulating the algorithm. MOST RELIABLE is a non-linear task so a non-linear encoder is expected to help. We also note that as the graphs grow in size, the number of reachable nodes in the priority queue increases, making it more likely we pick the wrong node without affecting the correctness of prediction.

Secondly, we note that in the NA setting (Tab. 2) the NE is able to solve the *Pred.* prediction quite well up to 100 nodes, but clearly found an alternative way of reasoning as the key prediction is hugely wrong for larger graphs. Note that the largest shortest distances will be found in 2-grid graphs, where it will be upper bounded by 51. Also for MOST RELIABLE the performance on *Pred.* drop at 50 nodes is significantly more severe with less representation power.

**Transfer yields little improvement**: The two key experiments are the transfer setting (§ 4.3) and the multi-task setting (§ 4.4). We hypothesised that the standard transfer experiments (fine-tune and freeze) would not help systematic generalisation. None of the transfer methods (Tab. 3) help generalise either task significantly. In fact, they harm systematic generalisation in terms of *Pred.* prediction in all cases. The only benefit that can be observed is better generalisation on *Key* accuracy indicating that the network outputs are less extreme. The best transfer method is 2-Processor as we hypothesised in § 4.3, which improves *Key* prediction at the cost of harming *Pred.* accuracy.

**Multi-task helps systematic generalisation**: In the multi-task set-up (Tab. 4), several things occur: the *Key* prediction generalises even better and is predicting in a reasonable range given the longest

Table 5: 2-Proc. transfer pre-trained on PRIM, DIJKSTRA, & DFS for sequential and BFS & BELLMAN-FORD for parallel. Pre-training on several tasks does not improve classic transfer.

| Model | #Nodes | MOST RELIABLE (SEQ) | | MOST RELIABLE (PAR) | |
|---|---|---|---|---|---|
| | | Key | Predecessor | Key | Predecessor |
| | 20 | $0.00237 \pm 0.0003$ | $0.163 \pm 0.06$ | $0.0408 \pm 0.009$ | $0.227 \pm 0.07$ |
| NE ++ 2-Proc. | 50 | $62.9 \pm 20$ | $0.606 \pm 0.05$ | $0.161 \pm 0.1$ | $0.363 \pm 0.04$ |
| | 100 | $1.76e6 \pm 4e5$ | $0.758 \pm 0.07$ | $2.68 \pm 4$ | $0.448 \pm 0.08$ |

Table 6: No algorithm (par.): For transfer we report the results of the best method (§ 4.3). Pre-trained on BFS and WIDEST respectively. Reliance on intermediate steps is lower for this class of problems, but multi-task transfer of knowledge is still beneficial in terms of size-generalisation.

| Model | #Nodes | BELLMAN-FORD | | MOST RELIABLE PATH | |
|---|---|---|---|---|---|
| | | Key | Predecessor | Key | Predecessor |
| | 20 | $0.0182 \pm 0.02$ | $0.057 \pm 0.02$ | $0.018 \pm 0.002$ | $0.226 \pm 0.06$ |
| NE (NA) | 50 | $59 \pm 80$ | $0.164 \pm 0.1$ | $0.147 \pm 0.2$ | $0.327 \pm 0.02$ |
| | 100 | $1.98e6 \pm 3e6$ | $0.261 \pm 0.2$ | $10.4 \pm 10$ | $0.435 \pm 0.05$ |
| | 20 | $0.00253 \pm 0.002$ | $0.028 \pm 0.01$ | $0.00957 \pm 0.004$ | $0.145 \pm 0.06$ |
| NE++ (NA) | 50 | $0.226 \pm 0.3$ | $0.057 \pm 0.01$ | $0.0367 \pm 0.04$ | $0.171 \pm 0.03$ |
| | 100 | $196 \pm 300$ | $0.095 \pm 0.04$ | $120 \pm 200$ | $0.217 \pm 0.02$ |
| | 20 | $0.0386 \pm 0.02$ | $0.072 \pm 0.03$ | $0.0221 \pm 0.006$ | $0.237 \pm 0.06$ |
| NE (Transfer Fine-tune) | 50 | $25 \pm 40$ | $0.162 \pm 0.05$ | $0.332 \pm 0.4$ | $0.331 \pm 0.02$ |
| | 100 | $1.72e5 \pm 2e5$ | $0.242 \pm 0.1$ | $230 \pm 300$ | $0.402 \pm 0.03$ |
| | 20 | $0.0223 \pm 0.02$ | $0.062 \pm 0.03$ | $0.0131 \pm 0.003$ | $0.196 \pm 0.07$ |
| NE++ (Transfer 2-Proc.) | 50 | $0.666 \pm 0.7$ | $0.105 \pm 0.005$ | $3.04 \pm 4$ | $0.313 \pm 0.05$ |
| | 100 | $10.8 \pm 10$ | $0.168 \pm 0.05$ | $579 \pm 800$ | $0.411 \pm 0.1$ |
| | 20 | $0.0154 \pm 0.02$ | $0.034 \pm 0.01$ | $0.173 \pm 0.1$ | $0.346 \pm 0.02$ |
| NE (Multi-task) | 50 | $6.22 \pm 9$ | $0.051 \pm 0.004$ | $0.407 \pm 0.4$ | $0.362 \pm 0.03$ |
| | 100 | $1.53e3 \pm 1e3$ | $0.096 \pm 0.02$ | $0.615 \pm 0.6$ | $0.376 \pm 0.05$ |
| | 20 | $0.00353 \pm 0.004$ | $0.023 \pm 0.01$ | $0.00672 \pm 0.0005$ | $0.153 \pm 0.06$ |
| NE++ (Multi-task) | 50 | $0.0141 \pm 0.02$ | $0.03 \pm 0.006$ | $0.00805 \pm 0.002$ | $0.182 \pm 0.01$ |
| | 100 | $8.84 \pm 10$ | $0.13 \pm 0.1$ | $0.00971 \pm 0.002$ | $0.212 \pm 0.02$ |

shortest path in graphs of size 100. Further, NE in this setting has similar *Pred.* accuracy on DIJKSTRA compared to NA, NE++ benefitted from the inductive bias in terms of its *Pred.* accuracy on graphs of size 100 for DIJKSTRA. The results on MOST RELIABLE are significantly improved and NE++ achieves good levels of systematic generalisation in solving the task. NE interestingly worsens in its performance on 20 nodes, but maintains a stable *Pred.* accuracy on larger graphs. Demonstrating that the inductive bias from WIDEST prevents overfitting in distribution and improves the performance on larger graphs. Overall, the results validate our initial hypothesis that multi-task learning is the correct approach to transfer knowledge.

**Trying to extract shared subroutines does not help transfer**: Finally, we study to what extent the models are able to separate the common shared subroutines and the subroutines individual to each algorithm by training multiple algorithms with TF in a multi-task set-up together (Tab. 5). If the processor successfully captures only the shared subroutines, then we might expect the transfer results to be improve. We can see in Tab. 5 that multi-task pre-training does not significantly improve results and that multi-task learning with the target algorithm is still the best approach. However, one alternative explanation is that given a good processor, the encoder struggles to learn the expected encoding by the processor and thus performs poorly.

## 6.2 Parallel

Parallel algorithms are significantly easier than sequential ones due to their much shorter length and the lack of a central data-structure that needs to be learned to execute. This can be observed in the much higher performance in the NA setting (Tab. 6). Interestingly, it seems that in this setting expressivity was helpful for systematic generalisation, even in the BELLMAN-FORD setting.

**Transfer harms performance**: In Tab. 6 we show only the best transfer result, but as we can see this actually harms *Pred.* accuracy for NE++ for both algorithms, while producing roughly the same result for NE. In both cases, the results suggest that random initialisations are better than transfer ones. We think this may because the shared algorithmic knowledge of parallel algorithms is already inherently captured by GNNs as they apply message functions in parallel to each edge, which then only need to learn the `relax_edge` function. Pre-training on several algorithms did not help (Tab. 5).

**Multi-task only helps Key accuracy**: Similarly to sequential reasoning multi-task vastly outperforms transfer techniques and significantly improves *Key* prediction compared to NA, while keeping *Pred.* similar. Contrary to sequential reasoning the *Pred.* prediction is comparable between NA and multi-task. We think this is due to the shorter execution length providing less of an inductive bias for the target algorithm and the strong inductive bias of GNN architectures towards parallel algorithms. However, the access to more stable gradients due to the multi-task learning approach seems to help learning to some extent due to the improved *Key* predictions. Further, we observe that when the model has less capacity (NE on MOST RELIABLE PATH) multi-task is still able to improve systematic generalisation on *Pred.* at the cost of slightly worse in-distribution (20 nodes) performance.

### 6.3 Why transfer learning fails?

Transfer via freezing and/or fine-tuning clearly does not work as demonstrated by the results in Tab. 3. The fact that having two processors, one frozen and one to fine-tune, also does not help transfer is telling, because neither the fine-tuning process losing information, nor the rigidity of the network can be at fault. In other words, fine-tuning has the disadvantage that we lose the original weights and hence potentially lose information. Freezing weights significantly limits the weights that can be changed and thus making it harder to fit the data. However, the 2-processor approach suffers from neither problem and yet still does not work.[8] Thus, we hypothesis that the reason why transfer fails to work is that the initial weights of a similar algorithm are not near a good (as in generalising) minimum for the target algorithm, in fact the minimum is often worse than the minimum found from randomly initialised weights (see Tab. 2).

### 6.4 Why multi-task fairs better?

Multi-task on the other hand does not rely on the weights being near a good minimum, but instead enforces them to be the same for the processor. This is a very different way to use the base algorithm as an inductive bias. This inductive bias is successful because the final weights are from a minimum that systematically generalises (on at least one of the algorithms) with the additional constraint that it performs well on the second target algorithm. For transfer the initial weights might systematically generalise on the original task there is no guarantee that the final weights stem from a minima that systematically generalises.

## 7 Conclusion

We set out to investigate how systematic generalisation could be improved on algorithmic tasks when the intermediate steps of the algorithm are not available. Inspired by the success of transfer learning in domains such as CV and NLP, we investigated it's applicability to learning graph algorithms in this setting. We showed that standard transfer learning is inadequate to leverage algorithmic knowledge learned from intermediate steps to new algorithmic tasks. Further, we showed how multi-task learning can enable the successful transfer of inductive biases learned from other algorithms when intermediate steps are available, significantly improving systematic generalisation. The results are especially strong in the more difficult sequential reasoning domain. Moreover, we conclude that expressivity can hurt systematic generalisation if the task is too easy and intermediate supervision is available. This should be taken into account when choosing the model. These disadvantages disappeared when trying to learn algorithmic reasoning without intermediate steps in our multi-task set-up, in this setting NE++ always outperforms the simpler architecture. Both architectures can achieve systematic generalisation. *Limitations* of our work are that the results are specific to algorithms on static graphs. Furthermore, as the number of execution steps increases faster than linear in the number of nodes results are likely to worsen significantly.

This paper's contributions are fundamental in nature and thus, the societal impact of this paper is low and there are no associated ethical risks. Any benefits or risks stem from further advances in reasoning systems that may be in some form be based on this work.

---

[8]Experiment 1 (in the Supplementary material) shows that the information from a processor can be used and recovered.

## Acknowledgments and Disclosure of Funding

We would like to thank Meng Qu, Zhaocheng Zhu, and Zuobai Zhang for proof reading the manuscript prior to submission.

This project is supported by the Natural Sciences and Engineering Research Council (NSERC) Discovery Grant, the Canada CIFAR AI Chair Program, collaboration grants between Microsoft Research and Mila, Samsung Electronics Co., Ltd., Amazon Faculty Research Award, Tencent AI Lab Rhino-Bird Gift Fund and a NRC Collaborative R&D Project (AI4D-CORE-06). This project was also partially funded by IVADO Fundamental Research Project grant PRF-2019-3583139727.

Petar Veličković is a Research Scientist at DeepMind.

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
