# OpenReview forum: "How to transfer algorithmic reasoning knowledge to learn new algorithms?"
_NeurIPS.cc/2021/Conference — NeurIPS 2021 Poster_

### Official Review · Reviewer_AnUG · 2021-07-11

**Rating:** 6
**Confidence:** 3

**Summary:**

This paper discusses how to transfer the knowledge from known step-by-step algorithms to learn new algorithms with only input and output pairs. The authors reach the conclusion: transfer learning does not help, but multi-task learning helps with the systematic generalization.
The main contributions of this paper are:
1. A new benchmark for transferring algorithmic knowledge
2. Study on training and loss functions case by case for different graph algorithms and setup
3. Shows standard transfer learning technique fails on algorithm learning, but multi-task learning helps.

**Limitations And Societal Impact:**

Limitations:
1. What is the multi-task learning performance on non-linear algorithms, such as SCC?
2. What is the transferability between algorithm types, such as transferring PATH to SCC?
To improve:
3. Better description of NeuralExecutor and NeuralExecutor++.
4. Unify the evaluation metric of  Key and Predecessor, either both "lower is better" or "higher is better".

**Main Review:**

### Originality: 3/5
Although there are a few works studying how to utilize neural networks to solve program/algorithm generation, few of them study algorithmic knowledge transferability in a principled way. The results reveal transfer learning is non-trivial in algorithmic tasks, and multi-task is a better choice.

### Quality: 3/5
The authors present a solid study of algorithmic transfer learning. They focus on the graph algorithms and vary on the a) transfer methods b) algorithm type c)algorithm difficulty d) graph size. The authors also note that all of these algorithms they have studied are O(n) and discuss the potential to extend to NP-hard problems. However, I still think adding polynomial complexity algorithms will help to make this work more complete, such as SCC. Further, the authors make an assumption that the algorithms are similar enough. Is it possible to transfer knowledge between different types of algorithms? For example, we know PATH is linear and is the prerequisite for learning SCC. Multi-task learning will probably run into a bottleneck in this setup.

### Clarity: 3/5
The overall structure and experiment setup are pretty clear. The description of NeuralExecutor and NeuralExecutor++ is a bit abstract, and it would be nice to illustrate them in a running example. It would also be helpful to have the evaluation metric of Key and Predecessor in a unified manner, either both "lower is better" or "higher is better".

### Significance: 3/5
The result reveals the current transfer learning techniques cannot be applied to algorithmic tasks easily. While multi-task learning can help with generalizability, this work may inspire future work on algorithmic task-specific transfer learning.

----------------------------------------------------------------------------------------------------------------------------------------------------------------------------------------
Update:

I have read the response from the authors and the other reviewers. The authors promise to improve the presentation and provide more experiments between different classes of algorithms. This helps to address my concern. On the other hand, the authors provide an intuitive analysis on why transfer learning fails to work on algorithm tasks, while multi-task learning can. The study is quite informative and I hope there will be a theoretical section analysis of the difference in the updated paper. In terms of the score, I would like to maintain my original assessment.

**Time Spent Reviewing:**

12 hours

---

> ### Author Response · Authors · 2021-08-11
> **Response to reviewer AnUG**
>
> We thank the reviewer for their detailed comments!
>
> **What is the multi-task learning performance on non-linear algorithms, such as SCC?**
>
> We assume SCC stands for strongly connected component, please correct us if we are wrong. To our understanding SCC is a linear algorithm as executed by a GNN (a GNN can executed O(|E|) operations at each step) because SCC can be done in a single depth first search (DFS) pass and DFS can be done in O(|V|). Apologies if we misunderstood.
>
> We considered trying bipartite matching as a more difficult problem, however, it's complexity is O(EV) and thus also only uses O(V) steps when executed by a GNN. Unfortunately NP-hard problems have exponential run-time and so would make checking size generalisation computationally infeasible.
>
> **What is the transferability between algorithm types, such as transferring PATH to SCC?**
>
> We assume algorithm types refers to parallel versus sequential reasoning. We briefly tested this early on in the research process to verify that indeed algorithms do need to be similar. We will run an experiment to test transfer between parallel and sequential algorithms.
>
> **Better description of NeuralExecutor and NeuralExecutor++.**
>
> We will revise the paper to clarify how different components (encoder, decoder, processor, etc.) are connected in each setting with a running example. However, we are sadly not able to update the pdf during the discussion period.
>
> **Unify the evaluation metric of Key and Predecessor, either both "lower is better" or "higher is better".**
>
> We will change the metric from accuracy to error rate to ensure that lower is always better. Thank you for the suggestion!

---

### Official Review · Reviewer_ZoqR · 2021-07-15

**Rating:** 4
**Confidence:** 4

**Summary:**

The authors studied how to transfer algorithmic reasoning knowledge. Particularly, they aim at using algorithms for which we have access to the execution trace to learn to solve similar tasks for which execution trace is not available. The authors examined two classes of graph algorithms, parallel algorithms and sequential greedy algorithms. They tested the hypothesis that standard transfer techniques are not sufficient and showed that multi-task learning can be used instead to achieve the transfer of algorithmic reasoning knowledge.


**Limitations And Societal Impact:**

The authors mentioned that the potential risk of having any negative societal impact is low and there are no associated ethical risks, which I largely agree with.

**Main Review:**

The work is relatively innovative in the sense that it addressed the transfer learning of algorithmic reasoning when intermediate steps are not available. It extended NeuralExecutor by Velicˇkovic ́ et al. so that the multi-task learning paradigm works even when execution traces are not available.

I have several concerns about the soundness of the paper. First, the authors' argument that NE++ is definitely better seems not conclusive. In Line 248 in the main text, it says that NE++ actually has worse performance in some cases (Tab. 1). Even with the help of multi-task training, the NE++’s Predecessor accuracy on DIJKSTRA nodes 100 (0.718, Tab. 4) is still worse than the NE’s Predecessor accuracy on DIJKSTRA nodes 100 (0.806, Tab. 2). Furthermore, in line 296 - 299, the authors mentioned that “Multi-task only helps improve Key accuracy, but not Pred. prediction".

There is another flaw with the paper. The authors state that “Key accuracy measures the node features mean squared error for all algorithms” (line 236-237), And the authors report the average with standard deviation in the main paper. I noticed that in some cases the mean and the standard deviation are extremely large (e.g., the cell of NE++(NA), nodes100, BELLMAN-FORD, key 1.98e6 ± 3e6). This is worrisome since an extremely large standard deviation indicates the instability of the results, making the corresponding conclusion less convincing. If the authors could respond to this that would be great.

The submission is not very clearly written. The tables lack proper caption to describe the information presented in the table. For instance, in table 1, the authors should briefly mention the definition of the evaluation metrics presented in the table. Otherwise, the readers won’t know how to interpret the numbers. Similarly, since each table represents a slightly different training paradigm, the caption should briefly capture the overall training schemes. Naming the table as “No algorithm (seq.)” is very confusing and not readable at all. Finally, the authors could consider highlighting the winning algorithms in bold in each training condition so that it's easier to compare the performances of multiple models.

------------------------

Updated: I've read the authors' responses to my review as well as the ones to the other reviewers'. I don't think my concerns were fully addressed. If the authors did not intend to argue that NE++ always works better than the NE model, then they need to clarify when NE++ is expected to be better than NE and why. It did not bring any insight into the problem if the authors only show the empirical results that NE++ "sometimes" works better than NE but sometimes doesn't. Also, although the authors re-stated the potential reason for the large standard deviation, it didn't address my concern. When the reported metrics have large standard deviations, the reported mean is less convincing and is likely driven by extreme values. Therefore, I'm not confident of the comparisons between those means. Given the above reasons, I decided to keep my original rating for the paper.


**Time Spent Reviewing:**

10 hours

---

> ### Author Response · Authors · 2021-08-11
> **Response to reviewer ZoqR**
>
> We thank the reviewer for their detailed comments!
>
> **I have several concerns about the soundness of the paper. First, the authors' argument that NE++ is definitely better seems not conclusive. In Line 248 in the main text, it says that NE++ actually has worse performance in some cases (Tab. 1). Even with the help of multi-task training, the NE++’s Predecessor accuracy on DIJKSTRA nodes 100 (0.718, Tab. 4) is still worse than the NE’s Predecessor accuracy on DIJKSTRA nodes 100 (0.806, Tab. 2).**
>
> We did not mean to claim that NE++ was always better than NE. We apologise for the confusion. But the paper is sound as we build NE++ to be able algorithmically align with a larger set of graph algorithms and NE++ is able to achieve this as it performs better on the non-linear relaxation graph algorithm (most reliable path), while maintaining good generalisation on the easier tasks. The fact that NE++ has sometimes worse performance on the linear relaxation graph algorithm Dijkstra is line with other research [1,2], which has also shown that the simplest architecture that aligns with the given task is able to do best. The problem is that as the neural networks grow more expressive they are better at finding “shortcuts” on the training data which do not generalise. This is in line with results in Tab. 1 as NE++ is able to outperform NE on the graphs of the original training size. We will clarify the paragraph starting line 248 further.
>
> [1] What can neural networks reason about? (Xu et al. ICLR’2020)
>
> [2] How Neural Networks extrapolate: From feedforward to graph neural networks (Xu et al. ICLR’21)
>
> **Furthermore, in line 296 - 299, the authors mentioned that “Multi-task only helps improve Key accuracy, but not Pred. prediction".**
>
> Indeed, one of the surprising results of this paper is that for parallel algorithms, which tend to have very short execution traces (equal to the diameter of the graph), the performance without intermediate steps is already excellent (NE++ > 0.9 on graphs of size 100 Tab. 6). However, predicting the predecessor in Dijkstra is a much easier metric to do well on than the key accuracy, as the predecessor
> We don’t believe that this affects the soundness of the paper, as it was trying to establish how transfer could be done. In the sequential setting the benefits of multitask are clearly shown.
>
> **There is another flaw with the paper. The authors state that “Key accuracy measures the node features mean squared error for all algorithms” (line 236-237), And the authors report the average with standard deviation in the main paper. I noticed that in some cases the mean and the standard deviation are extremely large (e.g., the cell of NE++(NA), nodes100, BELLMAN-FORD, key 1.98e6 ± 3e6). This is worrisome since an extremely large standard deviation indicates the instability of the results, making the corresponding conclusion less convincing. If the authors could respond to this that would be great.**
>
> The large standard deviation comes from the fact that the graph types we test on are quite different. We mentioned this in line 227 in the original paper, but we should have and will make this clearer when discussing the results as well. The results separated by graph type can also be found in the supplementary material (A.3).
>
> **The submission is not very clearly written. The tables lack proper caption to describe the information presented in the table. For instance, in table 1, the authors  should briefly mention the definition of the evaluation metrics presented in the table. Otherwise, the readers won’t know how to interpret the numbers. Similarly, since each table represents a slightly different training paradigm, the caption should briefly capture the overall training schemes. Naming the table as “No algorithm (seq.)” is very confusing and not readable at all. Finally, the authors could consider highlighting the winning algorithms in bold in each training condition so that it's easier to compare the performances of multiple models.**
>
> Yes, we agree. Due to space constraints we kept the captions minimal and explained the Tables in the main text, this was in hindsight a poor trade-off to make. We will expand the captions to include a brief explanation of the experiment and the key numbers to look for, we will bold the best answer, and change the metric from accuracy to error rate to ensure that lower is always better.
>
> We hope to have addressed all the concerns of the reviewer.

---

### Official Review · Reviewer_JA8z · 2021-07-17

**Rating:** 5
**Confidence:** 3

**Summary:**

The paper studies a transfer learning setting for learning graph algorithms, where we do not have the execution traces for the target algorithm, but we have execution traces for a related algorithm. The paper uses an adapted version of NeuralExecutor (Velickovic et al., 2019) and investigates transfer learning within two families of graph algorithms: sequential algorithms and parallel algorithms. The paper finds that standard transfer learning techniques don’t help in this setting, but multi-task training is useful.

**Ethics Review Area:**

["I don’t know"]

**Limitations And Societal Impact:**

The authors have addressed the limitations and potential negative societal impact. I do not have anything to add.

**Main Review:**

I am leaning towards rejecting the paper. My main concerns are the potential impact and technical novelty.

Strengths:
* The experiments are extensive and cover many different settings.
* The experiments have some interesting results. It is interesting to see that transfer learning and multi-task learning have quite different results.

Weaknesses:
* I am doubtful about the usefulness of this setting. It’s unclear what real-world scenarios require this type of transfer learning --- I would assume that execution traces are easy to get for graph algorithms (given their algorithmic nature).
* The technical contribution is a little thin. While the paper is the first to study the proposed transfer learning settings, the techniques used are not new (standard transfer/multi-task learning methods). The paper does propose a modified version of NeuralExtractor, so there is some novelty there.
* While the comparison between multi-task and transfer learning is interesting, the paper does not explain *why* the two methods are different very well. The paper provides some intuition in the introduction (line 44-53), but I find it a little vague (or maybe I didn't fully understand this part). I think this is an important takeaway from the paper, so maybe it's worth expanding this intuition with a more detailed example from experiments.

Suggestions:
* I would love to see more convincing examples of why this setting is important. It's possible that I miss some interesting applications, so I would be happy to upgrade my scores if this can be addressed.
* Another way to improve the paper is to provide a deeper (possibly theoretical) analysis for why transfer learning and multi-task learning are so different in the experiments. This will shed light on the difference between the two techniques and may inspire future research in both graph algorithm learning and multi-task/transfer learning.

Other minor suggestions/questions:
* The current draft seems to assume that the reader knows the concept of algorithmic alignment. The paper may be more accessible if there is an explanation of this concept in the background section.
* Is there a reason why L1 loss is used instead of L2?

**Update after author response:**

The author response clarifies the motivation for the setting, so I've upgraded my rating from 4 to 5. I recommend adding an expanded version of this discussion to the paper, since motivation is very important when proposing a new research direction. It would be even better if the discussion can be connected to the particular graph algorithms used in the experiments.

However, I am still leaning towards rejection. The author response does not provide a convincing explanation of why multi-task learning and transfer learning, so I think the empirical analysis can still be improved.

**Time Spent Reviewing:**

3

---

> ### Author Response · Authors · 2021-08-11
> **Response to reviewer JA8z**
>
> We thank the reviewer for their detailed comments!
>
> **I am doubtful about the usefulness of this setting. It’s unclear what real-world scenarios require this type of transfer learning --- I would assume that execution traces are easy to get for graph algorithms (given their algorithmic nature).**
>
> Indeed, for any known graph algorithm we could get the execution trace. However, there are many cases where we know that the answer to a problem is computed using something that is either a graph algorithm or similar to one. One such example is knowledge graph link prediction. There exists a reasoning process that can answer the question: what is the tail entity given the head entity and the relation. However, we do not have access to its execution trace, i.e. the step-by-step deductions of the reasoning process, but we do have input-output pair examples. It is highly likely that the reasoning process is at least similar to a graph algorithm given that both share similar generalisation properties and operate on a graph as input. We envision that the set-up studied and the direction proposed in this paper may later help learn neural networks that can answer knowledge graph link prediction problems and be able to generalise their reasoning to new graphs and larger graphs. Doing so in this paper, however, is outside the scope. A similar scenario may be encountered in physics, where given a set of particles we would like to simulate their behaviour. We may know the final outcome as it has been determined experimentally, but we are unlikely to have access to the intermediate states of the particle.
>
> A slightly different, but related application may be when the data the graph algorithm needs to operate on is encoded in a high-level space. This is a scenario encountered in reinforcement learning and [1] has already started investigating the process of pre-training an encoder with graph algorithms.
>
> [1] XLVIN: eXecuted Latent Value Iteration Nets (Deac et al. 2020)
>
> **The technical contribution is a little thin. While the paper is the first to study the proposed transfer learning settings, the techniques used are not new (standard transfer/multi-task learning methods). The paper does propose a modified version of NeuralExtractor, so there is some novelty there.**
>
> The technical contribution of this paper are to give a recipe as to how transferring algorithmic knowledge (multi-task with the simplest model that still aligns with tasks is best), we propose a more expressive variant of Neural Executor, and we give several tricks as to how to train them (e.g. using the max of several trajectories line 56 & 176). However, we believe the majority of the contribution of this paper lies in proposing the research direction, expanding existing benchmarks, and demonstrating that standard wisdom with regards to transfer does not apply to algorithmic reasoning.
>
> **While the comparison between multi-task and transfer learning is interesting, the paper does not explain why the two methods are different very well. The paper provides some intuition in the introduction (line 44-53), but I find it a little vague (or maybe I didn't fully understand this part). I think this is an important takeaway from the paper, so maybe it's worth expanding this intuition with a more detailed example from experiments.**
>
> Yes, we agree and we will expand on this in the revised draft.
>
> Transfer via freezing and/or fine-tuning clearly does not work as demonstrated by the results in Tab. 6. The fact that having two processors, one frozen and one to fine-tune, also does not help transfer is telling, because neither the fine-tuning process losing information, nor the rigidity of the network can be at fault. In other words, fine-tuning has the disadvantage that we lose the original weights and hence potentially lose information. Freezing weights significantly limits the weights that can be changed and thus making it harder to fit the data. However, the 2-processor approach suffers from neither problem. This reduces the reasons as to why transfer learning might fail significantly. The fact that multi-task learning is able to help shows that there is information that can be reused to improve performance on the target task. Hence, we can confirm that the training data of the base algorithm with intermediate steps contains useful information, but that transfer learning is not able to extract such information from the weights. While this does not allow us to definitively answer why transfer doesn’t work, we can exclude the most common explanations.
>
> One experiment we will try to run during the discussion period to further narrow down why is to check whether a pre-trained processor can be used to train a new set of encoder and decoders on training data without intermediate steps on the original task. A positive answer would imply that the information is indeed contained in the processor, but that extracting the transferable aspect it is highly difficult, something that would come naturally to multi-task learning.
>
> **The current draft seems to assume that the reader knows the concept of algorithmic alignment. The paper may be more accessible if there is an explanation of this concept in the background section.**
>
> You are correct we should have explained the concept of algorithmic alignment in more detail and will add the following explanation to the main paper if there is space or else in the appendix:
>
> Algorithmic alignment [2] refers to the concept of a computation structure---in our case neural networks---and the structure of an algorithm 'aligning'---in this paper graph algorithms. 'Alignment' means there exists a mapping between substructures of the computation architecture and simple substeps of the algorithm, where the substructures can 'easily' compute the corresponding substep they are mapped to---for some definition of easy, usually linear. This means that the substructures can learn these 'easy' subfunctions instead of having to coordinate with other parts of the architecture to recreate the highly non-linear function represented by the complete algorithm.
>
> [2] What can neural networks reason about? (Xu et al. ICLR’2020)
>
> **Is there a reason why L1 loss is used instead of L2?**
>
> L2 loss was prone to growing very large and dominating the other losses on predecessor prediction or next node prediction. L1 loss behaved better in those scenarios and didn’t lead to performance in other settings either.

---

### Official Review · Reviewer_A7vb · 2021-07-22

**Rating:** 6
**Confidence:** 3

**Summary:**

The paper studies the problem of transferring algorithmic reasoning knowledge from a task to another similar task. The authors first propose a set of benchmark tasks to evaluate transfer learning for algorithmic reasoning on graphs.

The authors then propose modifications to the Neural Extractor model by replacing the node encoder with a latent edge encoder. Along with this, they propose using two latent encoders, one with a linear layer and one with a non-linear layer to allow the model to learn more types of tasks. Also, a separate encoder is used for each task.

The authors experiment with several transfer-learning and Multi-task learning (MTL) approaches on the proposed benchmark tasks.
The authors conclude that standard methods for transfer learning are ineffective on transferring algorithmic reasoning knowledge. They further show that the MTL approach effectively transfers knowledge thus improving systematic generalization.


**Limitations And Societal Impact:**

The authors address the limitations and societal impact of the approach.

**Main Review:**

Strengths:
1. The paper presents a set of tasks to benchmark transfer learning of algorithmic reasoning. This is an important problem and the proposed tasks would help further research in this area.
2. The paper proposes modifications to the neural executor model that improves the performance on certain metrics
3. The results and conclusions drawn by the paper about transfer learning methods are interesting and would be helpful to the community for future research in this area.


Concerns, questions, and suggestions
1. An analysis of the results is lacking. As mentioned, the results lead to some interesting conclusions, such as pertaining methods are ineffective compared to multi-task learning. The paper might benefit from some experiments that shed light on why this would be the case. Does some sort of catastrophic forgetting occur when the model is fine-tuned on a different task? And could this be better illustrated with an experiment?
2. The results are hard to understand in the way they are presented. The reader is often required to compare numbers from different tables on different pages. Perhaps, a graph comparing the results would be more intuitive. If possible, the authors can add (in the caption), what the reader needs to look for in the results table.
3. In line 301, the authors claim that the gradients are more stable due to MTL. Could the authors expand on this?
4. Line 153 - delete repeated has

Overall, I think the paper reaches an interesting result on the comparison of pretraining methods vs MTL for transfer learning for algorithmic reasoning. I feel the arguments of the paper would be bolstered by some analysis that better elucidates the reason for the claims.



**Time Spent Reviewing:**

4

---

> ### Author Response · Authors · 2021-08-11
> **Response to reviewer A7vb**
>
> We thank the reviewer for their detailed comments!
>
> **The paper might benefit from some experiments that shed light on why this would be the case. Does some sort of catastrophic forgetting occur when the model is fine-tuned on a different task? And could this be better illustrated with an experiment?**
>
> We agree that the paper’s discussion section could have used more analysis as to why we see the results we see.
>
> Catastrophic forgetting might be occurring in the transfer set-up with fine-tuning, but is unlikely to be the main culprit because catastrophic forgetting cannot occur in the two other transfer settings: Freeze and 2-Processor. In each of these settings the processor weights---the weights that are transferred---are frozen and thus cannot be forgotten by the network. Given that all transfer settings fail it is unlikely that catastrophic forgetting is the cause of the poor performance. This may lead to the hypothesis that the transferable information is stored in the decoder and encoder, however, this cannot be the case by design as they only operate on nodes without considering edge information and only the processor has the necessary expressivity and structure to generalise the training data. While this does not allow us to definitively answer why transfer doesn’t work, we can exclude many possible explanations.
>
> We will include these explanations of the results in the discussion section. Thank you for pointing out the gap in our discussion of the results!
>
> **The results are hard to understand in the way they are presented. The reader is often required to compare numbers from different tables on different pages. Perhaps, a graph comparing the results would be more intuitive. If possible, the authors can add (in the caption), what the reader needs to look for in the results table.**
>
> Yes, we agree. Due to space constraints we kept the captions minimal and explained the Tables in the main text, this was in hindsight a poor trade-off to make. We will expand the captions to include a brief explanation of the experiment and the key numbers to look for, we will bold the best answer, and change the metric from accuracy to error rate to ensure that lower is always better.
>
> **In line 301, the authors claim that the gradients are more stable due to MTL. Could the authors expand on this?**
>
> In the multi-task learning set-up the gradient consists of the two gradients for each task: the task with intermediate steps (the inductive bias) and the target task without intermediate steps. In general, teacher-forcing is going to yield less noisy gradients because there is no accumulation of errors. Thus, the combined gradient of a teacher-forcing gradient and the non-teacher-forcing gradient will be more ‘stable’ than only having the non-teacher-forcing gradient. We will expand on this in the main paper.
>
> **Line 153 - delete repeated has**
>
> Thank you for spotting this!

---

### Author Response · Authors · 2021-08-11
**Summary of the responses and changes**

The main areas of concern raised by the reviewers was to improve clarity of the results and architecture description as well as a more detailed explanation as to why transfer learning performs poorly.

**Changes to the draft**

 - Expand the introduction with a clearer example of the applicability of this direction. (Reviewer JA8z)
 - We will expand the captions, we will bold the best answer, and change the metric from accuracy to error rate to ensure that lower is always better (Reviewers A7vb, ZoqR, AnUG ).
 - We will revise the paper to clarify how different components (encoder, decoder, processor, etc.) are connected in each setting with a running example (Reviewer AnUG).
- Add a paragraph to explain the concept of algorithmic alignment (Reviewer JA8z).
- We will further expand the discussion of the results to more clearly explain why transfer learning might fail (Reviewers A7vb, JA8z).
- We will explain the reason for the sometimes large standard deviation in more detail in the results (Reviewer ZoqR).

**Further experiments**

These are experiments we look to run during the discussion period and will add results for as they become available.
- Check whether a pre-trained processor can be used to train a new set of encoder and decoders on training data without intermediate steps on the original task.
- Test transfer between parallel and sequential algorithms

We thank all the reviewers for their constructive suggestions and comments!

---

### Author Response · Authors · 2021-08-27
**More results and addressing the concerns of the reviewers**

**We first give the results of the experiment we said we would run and discuss the primary question that reviewers had which is why transfer fails. We finish by addressing the concern about the standard deviation of Reviewer ZoqR.**

Experiment 1:

Pre-train on Dijkstra with teacher forcing, transfer with a frozen processor to see to what extent the encoder/decoder can be recovered.

Transfer from Dijkstra to Dijkstra NeuralExec2 (Frozen)

| Node accuracy     | Key MSE           | Predecessor accuracy |
|-------------------|-------------------|----------------------|
| $0.886 \pm 0.03 $ | $0.226 \pm 0.09 $ | $0.977 \pm 0.002 $   |
| $0.543 \pm 0.03 $ | $3.51 \pm 4 $     | $0.917 \pm 0.03 $    |
| $0.241 \pm 0.02 $ | $132 \pm 200 $    | $0.774 \pm 0.1 $     |

Transfer from Dijkstra to Dijkstra NeuralExec3 (Frozen)

| Node accuracy     | Key MSE          | Predecessor accuracy |
|-------------------|------------------|----------------------|
| $0.89 \pm 0.01 $  | $0.253 \pm 0.1 $ | $0.922 \pm 0.03 $    |
| $0.381 \pm 0.06 $ | $4.35 \pm 4 $    | $0.656 \pm 0.02 $    |
| $0.161 \pm 0.03 $ | $35.1 \pm 40 $   | $0.443 \pm 0.09 $    |

Conclusion: The results are mostly quite similar to the orignal results, but slightly worse, thus indicating that while the re-use of a pre-trained processor is not trivial it is no the primary reason for transfer to fail.

Experiment 2: Transfer between algorithm classes

Transfer from BF to Dijkstra NeuralExec2 (2-Processor)

| Key MSE               | Predecessor accuracy |
|-----------------------|----------------------|
| $0.00653 \pm 0.005 $  | $0.908 \pm 0.04 $    |
| $1.69e4 \pm 2e4 $    | $0.599 \pm 0.05 $    |
| $1.11e12 \pm 2e12 $ | $0.229 \pm 0.2 $     |

Transfer from BF to Dijkstra NeuralExec3 (2-Processor)

| Key MSE               | Predecessor accuracy |
|-----------------------|----------------------|
| $0.000531 \pm 0.0003 $ | $0.95 \pm 0.03 $  |
| $12.1 \pm 10 $         | $0.751 \pm 0.1 $  |
| $74500 \pm 100000 $    | $0.498 \pm 0.07 $ |

Transfer from BF to Dijkstra NeuralExec3 (Finetune)

| Key MSE               | Predecessor accuracy |
|-----------------------|----------------------|
| $0.000192 \pm 0.0002 $ | $0.973 \pm 0.02 $ |
| $2.01e4 \pm 3e4$     | $0.33 \pm 0.1 $   |
| $1.25e14 \pm 2e14 $  | $0.284 \pm 0.05 $ |

Transfer from BF to Dijkstra NeuralExec2 (Finetune)

| Key MSE               | Predecessor accuracy |
|-----------------------|----------------------|
| $0.00029 \pm 0.0003 $ | $0.974 \pm 0.01 $ |
| $7.94e5 \pm 1e6 $   | $0.559 \pm 0.07 $ |
| $8.32e15 \pm 1e16 $ | $0.373 \pm 0.09 $ |

Transfer from BF to Dijkstra NeuralExec3 (Freeze)

| Key MSE               | Predecessor accuracy |
|-----------------------|----------------------|
| $0.000623 \pm 0.0004 $ | $0.959 \pm 0.02 $ |
| $79.2 \pm 90 $         | $0.63 \pm 0.09 $  |
| $1.23e6 \pm 2e6 $  | $0.138 \pm 0.08 $ |

Transfer from BF to Dijkstra NeuralExec3 (Freeze)

| Key MSE               | Predecessor accuracy |
|-----------------------|----------------------|
| $0.0108 \pm 0.007 $   | $0.904 \pm 0.05 $ |
| $152 \pm 200 $        | $0.664 \pm 0.01 $ |
| $3.29e9 \pm 4e9 $ | $0.478 \pm 0.09 $ |

Multitask BF (TF) and Dijkstra (NA) NeuralExec3

| Key MSE               | Predecessor accuracy |
|-----------------------|----------------------|
| $1.47 \pm 2 $         | $0.637 \pm 0.07 $ |
| $5.2e9 \pm 5e9 $  | $0.231 \pm 0.1 $  |
| $3.14e29 \pm 3e29 $ | $0.248 \pm 0.05 $ |

Multitask BF (TF) and Dijkstra (NA)  NeuralExec2

| Key MSE             | Predecessor accuracy |
|-----------------------|----------------------|
| $0.0994 \pm 0.1 $ | $0.871 \pm 0.08 $ |
| $0.926 \pm 1 $    | $0.845 \pm 0.1 $  |
| $3.77 \pm 4 $     | $0.788 \pm 0.1 $  |

Conclusion: Transfer learning did not work before and so it is not surprising it still does not work. Multitask learning does not work very well either and in both cases does worse than the NA baseline.

The primary concern of the reviewers is the lack of insight into why transfer does not work, but multi-task does. We apologise for not discussing this in more detail in the submitted paper, but we have significantly expanded the analysis on this question and will add it to the final paper. Please find a high level summary below:

**Why does transfer fail?**:

Transfer via freezing and/or fine-tuning clearly does not work as demonstrated by the results in Tab. 6. The fact that having two processors, one frozen and one to fine-tune, also does not help transfer is telling, because neither the fine-tuning process losing information, nor the rigidity of the network can be at fault. In other words, fine-tuning has the disadvantage that we lose the original weights and hence potentially lose information. Freezing weights significantly limits the weights that can be changed and thus making it harder to fit the data. However, the 2-processor approach suffers from neither problem and yet still does not work. Experiment 1 shows that the information from a processor can be used and recovered. Thus, given all of this information the most likely reason why transfer fails to work is that the initial weights of a similar algorithm are not near a good (as in generalising) minimum for the target algorithm, in fact the minimum is often worse than the minimum found from randomly initialised weights (see the no algorithm baselines).

Multitask on the other hand does not rely on the weights being near a good minimum, but instead enforces them to be the same. This is a very different way to use the base algorithm as an inductive bias. This inductive bias is successful because the final weights are from a minimum that systematically generalises (on at least one of the algorithms), while for transfer the initial weights might systematically generalise on the original task there is no guarantee that the final weights stem from a minima that systematically generalises.


**Regarding the standard deviation concern**:

We politely disagree with the reviewer. We think the standard deviation is to be expected given the widely different graph types. Given that all edge weights have expected value 0.6 the max expected shortest distance in a grid graph of size n is 0.6*(n/2+1), which is orders of magnitude larger than for an Erdos-Renyi or Barabasi-Albert graph, which will have a diameter of O(log(n)) and thus a max expected shortest distance of 0.6*log(n). This large difference in the shortest path makes large key errors when generalising much more likely on a grid-graph than on the other types of graphs. Vice versa is true for predecessor prediction: on a grid graph the degree of node is between 2 and 4 (constant no matter the size of the graph), while for a Barabasi-Albert and Erdos-Renyi graph it will grow with the size of the graph and be much larger, making errors much more likely. Again yielding vastly different error percentages. Taking this into account, we don't think the standard deviation is concerning but rather expected. We apologise for not being clearer in the main paper and will add this to the discussion.

**Finally, we would like to thank the reviewers for their interest and constructive criticism and comments helping to make this a better paper.**

---

### Decision · Program_Chairs · 2021-09-27

**Decision:**

Accept (Poster)

**Comment:**

This paper investigates how algorithms for which we have access to the execution trace can be leveraged to learn to solve similar tasks for which we do not have execution traces. The authors create a dataset that covers 9 algorithms and 3 different graph types to investigate transfer learning in two major classes of graph algorithm, parallel and sequential. They also introduce modifications to the existing NeuralExtractor model that improve its performance in some experimental settings.

The paper makes an interesting empirical discovery: according to their results, standard methods for transfer learning are ineffective in transferring algorithmic reasoning knowledge, while multi-task learning does effectively transfer this kind of knowledge. I think this observation is important to share with the community, particularly as it arises from experiments that Reviewer JA8z says "are extensive and cover many different settings." One possible concern with reliability of the finding is the large standard deviation in some of the results, as pointed out by Reviewer ZoqR.

A relevant issue raised by several reviewers is that the paper offered no analysis of why MTL is better than pre-training approaches for algorithmic transfer learning. In their response, the authors admit that they cannot provide a conclusive explanation for this observation, although they do argue for the exclusion of some standard hypotheses like catastrophic forgetting. They also conducted two more experiments during the review period that reveal more about why transfer learning might fail (although they don't prove their hypothesis on generalizing minima conclusively). However, I think it's fine for papers to identify important open questions and leave their answers as future work for the community.

Some reviewers questioned the paper's technical novelty (the modelling innovation amounts only to a modified NeuralExtractor that extends prior work), but I don't think that's what this paper is really about. Its impact lies in the novelty of its empirical observations.

Finally, the authors pledged to address reviewers' concerns around the clarity of the results presentation and the detail of their model description. I'm confident that they can do this straightforwardly.

For these reasons, I lean towards accepting the paper even though its aggregate review score puts it slightly below the acceptance threshold.